# Characterization of HLA-A/HLA-B/HLA-C/HLA-DRB1 Haplotypes in Romanian Stem Cell Donors Through High-Resolution Next-Generation Sequencing

**DOI:** 10.3390/ijms26115250

**Published:** 2025-05-29

**Authors:** Andreea Mirela Caragea, Radu-Ioan Ursu, Laurențiu Camil Bohîlțea, Paul Iordache, Alexandra-Elena Constantinescu, Ileana Constantinescu

**Affiliations:** 1Department of Immunology and Transplantation Immunology, “Carol Davila” University of Medicine and Pharmacy, 050474 Bucharest, Romania; andreea.caragea@drd.umfcd.ro (A.M.C.); alexandra-elena.constantinescu0720@stud.umfcd.ro (A.-E.C.); ileana.constantinescu@imunogenetica.ro (I.C.); 2Center for Immunogenetics and Virology, Fundeni Clinical Institute, 050474 Bucharest, Romania; 3Department of Medical Genetics, “Carol Davila” University of Medicine and Pharmacy, 050474 Bucharest, Romania; 4Department of Epidemiology, “Carol Davila” University of Medicine and Pharmacy, 050474 Bucharest, Romania; paul.iordache@umfcd.ro; 5“Emil Palade” Center of Excellence for Young People in Scientific Research (EP-CEYR), Academy of Romanian Scientists, AORS, 050045 Bucharest, Romania

**Keywords:** HLA haplotype frequencies, immunogenetics, population genetics, next-generation sequencing (NGS)

## Abstract

Human Leukocyte Antigen (HLA) genes are remarkable for their structural complexity and polymorphism. Located on chromosome 6 within the Major Histocompatibility Complex (MHC), these genes exhibit significant frequency variations across human populations and play a crucial role in immune responses, disease susceptibility, and transplant compatibility. This study aimed to assess the genetic profiles and HLA-A/HLA-B/HLA-C/HLA-DRB1 haplotype frequencies in a Romanian cohort. Whole venous blood samples were collected from 405 healthy, unrelated Romanian volunteers. Using next-generation sequencing (NGS), the study population was genotyped for HLA class I (HLA-A, HLA-B, and HLA-C) and class II (HLA-DRB1) loci. Haplotype frequencies were estimated using the expectation-maximization algorithm, addressing phase and allelic ambiguity. The Romanian cohort was compared with multiple populations sourced from the Allele Frequencies Net Database. The study identified 635 different HLA-A/HLA-B/HLA-C/HLA-DRB1 haplotypes. Among them, two haplotypes had frequencies close to 3%: HLA-A*01:01:01/HLA-B*08:01:01/HLA-C*07:01:01/HLA-DRB1*03:01:01, with a frequency of 3.33%, and HLA-A*02:01:01/HLA-B*18:01:01/HLA-C*17:01:01/HLA-DRB1*11:04:01, with a frequency of 2.84%. All other 633 haplotypes (approximately 99.7% of the total) had frequencies below 1%. The results of the current study underscore the extremely high diversity of HLA haplotypes in this population and the fact that even the most frequent haplotypes are relatively low in prevalence (each under 5% in this cohort). These findings and the great haplotypical diversity detected highlight the importance of NGS and high-resolution HLA typing in hematopoietic stem cell and solid organ transplantation, while also contributing to the better understanding of the area-specific population genetics resulting from historical regional dynamics. Further research with larger cohorts is necessary to validate these findings and expand upon their clinical implications.

## 1. Introduction

The human leukocyte antigen (HLA) system, encoding for major histocompatibility complex (MHC) proteins, plays a fundamental role in the adaptive immune system by presenting peptide antigens to Tcells, thus initiating immune responses. The extended variation in the HLA genes, particularly within the HLA-A, HLA-B, HLA-C, and HLA-DRB1 loci, results from historical selective pressures exerted by pathogens, environmental factors, and population dynamics [1]. This diversity is crucial not only in understanding human immune functions, but it also provides significant insights into population genetics, evolutionary biology, and the study of ancient human migrations [2].

Eastern Europe, comprising the Balkans, the Caucasus, and the Black Sea-adjacent countries, is particularly interesting for HLA genetic research due to its strategic historical position as a crossroads between Asia, Europe, and the Middle East. This region’s HLA diversity reflects the influence of various migration events, from ancient Indo-European migrations to more recent movements during the Ottoman and Byzantine Empires [3,4]. Studies on the genetic makeup of populations in Bulgaria, Macedonia, and Serbia reveal distinctive HLA haplotypes that show both local adaptations and genetic continuity with Near Eastern populations, suggesting a complex interplay of isolation and gene flow [5,6].

Research comparing HLA haplotypes between neighboring European countries, such as Greece, Turkey, and Albania, reveals similarities consistent with ancient population mixing during the Neolithic expansion, likely due to the introduction of farming practices from the Fertile Crescent [7,8]. Such findings support the hypothesis that the spread of agriculture significantly influenced the genetic landscape of Europe, bringing with it a suite of HLA alleles beneficial for immunity in diverse environments [9].

Comparative genetic studies have further elucidated the relationships between Eastern European populations and other groups. The HLA profiles observed in populations of Croatia and Hungary, for example, show genetic affinities with both Central European and Balkan neighbors, while the Slavic populations of Ukraine and Russia display a significant divergence from Western European gene pools [10,11]. These findings align with historical records of migration patterns, suggesting that genetic markers such as HLA haplotypes can effectively track human movements and inter-population relationships over millennia [12,13].

Thus, the study of HLA haplotypes provides a unique lens through which to examine the historical and biological narratives of Eastern European populations. By investigating the HLA genetic landscape, researchers gain insights into the ancient migratory paths, disease resistance profiles, and transplantation compatibility that shape these communities today. Expanding knowledge on HLA diversity not only enriches our understanding of population genetics but also has profound implications for fields as varied as immunology, evolutionary biology, and clinical transplantation.

From a clinical perspective, HLA diversity is essential in medical fields such as immunogenetics, organ transplantation, and disease association studies (e.g., HLA-B*08 in certain Eastern European populations and the increased susceptibility to autoimmune diseases, including multiple sclerosis and celiac disease, or the crucial importance of HLA haplotypes—especially including the HLA-A, HLA-C, and HLA-DRB1 loci—in heart transplantation) [14,15]. These associations highlight the role of HLA variants in both population health and personalized medicine, offering insights into genetic predispositions that vary across ethnicities and regions [16].

Moreover, studies have shown that regional HLA profiles impact the success of hematopoietic stem cell and organ transplants. The distribution of HLA haplotypes among potential donors and recipients within Eastern Europe is of paramount importance, as matching rates are generally higher among ethnically similar individuals [17,18]. Eastern European countries, with their unique HLA haplotype patterns shaped by historical admixture, underscore the need for expanding donor registries to reflect regional genetic diversity accurately, thereby improving transplantation outcomes [19,20].

Next-generation sequencing (NGS) has revolutionized modern diagnostics by providing a powerful tool for identifying genetic variations with high accuracy and resolution. In clinical practice, NGS is widely used for genetic disease diagnosis and cancer mutation profiling, complementing conventional PCR-based methods in the past decade [21]. This technology allows for the sequencing of vast amounts of genetic material quickly and efficiently, making it a key method for stem cell donor matching, particularly in HLA typing where high-resolution analysis is essential for reducing transplant rejection and improving patient outcomes. NGS enables the discovery of rare genetic variants that traditional methods may miss, thus enhancing our understanding of genetic diseases and providing more personalized treatment strategies.

The main goal of the current study is to detect the frequencies of HLA-A/HLA-B/HLA-C/HLA-DRB1 haplotypes in the analyzed Romanian cohort. The research also aims to underline the importance of using high-resolution NGS typing in HLA testing and to emphasize its utmost importance especially in stem cell and organ transplantation, by presenting, separately, the results at 2-, 4- and 6-digits resolution.

This study seeks to provide a novel contribution to the understanding of HLA haplotype frequencies in the Romanian population by utilizing high-resolution NGS. Previous studies of HLA distributions in Romania were primarily conducted at lower resolution or in specific regions/disease cohorts, whereas our approach offers a comprehensive nationwide perspective at the highest allelic resolution [20].

## 2. Results

HLA Allele Frequencies: In total, we detected 35 distinct HLA-A alleles, 59 HLA-B alleles, 29 HLA-C alleles, and 27 HLA-DRB1 alleles in the cohort. For example, HLA-A*02:01:01 was the most common HLA-A allele (~26%), followed by HLA-A*01:01:01 (~22%). The most frequent HLA-B allele was B*35:01:01 (~14%), while for HLA-C the highest frequency was observed for C*07:01:01 (~20%). The DRB1 locus was dominated by the alleles DRB1*11:01:01 (~15%) and DRB1*16:01:01 (~12%) [22,23].

Table 1 reveals the top 5 HLA-A, HLA-B, HLA-C, HLA-DRB1 alleles and their respective allele frequencies [22,23].

### 2.1. HLA Haplotypes: 6-Digits Resolution

The results of the study revealed a number of 635 different HLA-A/HLA-B/HLA-C/HLA-DRB1 haplotypes. Table 2 presents the 16 most frequent HLA haplotypes (6-digits) observed in the cohort.

Out of all the detected haplotypes, two revealed frequencies close to 3%: HLA-A*01:01:01/HLA-B*08:01:01/HLA-C*07:01:01/HLA-DRB1*03:01:01, HF 3.33%, identified in 27 individuals, and HLA-A*02:01:01/HLA-B*18:01:01/HLA-C*17:01:01/HLA-DRB1*11:04:01 (2.84%, 23 individuals) (Table 2).

Apart from these two haplotypes, all the other (approx. 99.7% of all resulted haplotypes) had frequencies lower than 1%, being detected in six participants or less each, of which 14 with HFs of 0.49–0.74% (4–6 individuals each) and the other 619 being detected in just one, two, or three participants each (rare haplotypes, HF < 0.49%). Two of the fourteen haplotypes with frequencies of 0.49–0.74% were identified in six individuals each (HF 0.74%), three in five persons each (HF 0.62%), and nine in four (HF 0.49%) (Table 2, Figure 1).

Of the 619 rare HLA haplotypes, the great majority (556 haplotypes) were observed in just one individual each (HF 0.12%), while 48 were detected in two individuals each (HF 0.25%), and 15 in three (HF 0.37%).

Combining the frequencies of all detected HLA haplotypes with frequencies lower than 0.49% (rare combinations), the calculated HF would reach 97.5%, most of these being represented by the combined frequency (87.6%) of all haplotypes detected just once in our cohort (all haplotypes with HF of 0.12%).

### 2.2. HLA Haplotypes: 4-Digits Resolution

When combining all the 6-digit sub-allele variants into 4-digit allele names for each locus, a similar total of 630 different HLA-A/HLA-B/HLA-C/HLA-DRB1 haplotypes was distinguished. Table 3 lists the most frequent 16 haplotypes at a 4-digit resolution.

Two different haplotypes showed the highest 4-digit haplotype frequencies: HLA-A*01:01/HLA-B*08:01/HLA-C*07:01/HLA-DRB1*03:01 and HLA-A*02:01/HLA-B*18:01/HLAC*07:01/HLA-DRB1*11:04, which were detected in 27 (HF 3.33%) and 23 individuals (HF 2.84%), respectively (Table 3, Figure 2).

Apart from these two haplotypes, none of the others revealed HF higher than 0.74%; all the remaining 628 haplotypes (~99.7% of all 4-digit haplotypes) were identified in six or fewer individuals. Two of the fifteen haplotypes with frequencies between 0.49% and 0.74% were detected in six persons each (HF 0.74%), three in five persons each (HF 0.62%), and 10 in four persons each (HF 0.49%) (Table 2). The remaining 612 haplotypes were rare (identified in just one, two, or three participants each, HF < 0.49%). In aggregate, haplotypes with HF < 1% accounted for ~94% of the total frequency. A total of 548 of these rare 4-digit haplotypes were observed in just one individual each (HF 0.12%), while 50 were detected in two individuals (HF 0.25%), and 15 in three individuals (HF 0.37%).

Combining the frequencies of all detected HLA haplotypes with frequencies lower than 1%, the calculated HF would reach approx. 87% (the combined frequency of all haplotypes detected just once in our cohort—HF 0.12%), emphasizing again the absence of any high-frequency haplotype in this population.

### 2.3. HLA Haplotypes: 2-Digit Resolution

Merging all the sub-variants of the main HLA class I alleles into 2-digit HLA-A/HLA-B/HLA-C/HLA-DRB1 haplotypes, 576 different haplotypes have been identified. Table 4 presents the top 24 2-digit haplotypes detected in the analyzed cohort.

The results revealed the same top two haplotypes, but with slightly higher frequencies, being observed in one individual more each than in the 4-digits and 6-digits analysis: HLA-A*01/HLA-B*08/HLA-C*07/HLA-DRB1*03 (28 individuals, HF 3.46%) and HLA-A*01/HLA-B*18/HLA-C*03/HLA-DRB1*11 (2.96%, 24 individuals) (Table 4, Figure 3).

All the remaining 574 haplotypes (99.65%) revealed frequencies of 0.74% or lower (detected in at most six participants each). Of these, 22 showed frequencies ranging from 0.49% and 0.74% (three with HF 0.74%, eight with HF 0. 62% and eleven with HF 0.49%) (Table 4), while the other 552 were rare haplotypes (identified in one, two, or three individuals each; HFs of 0.12% to 0.37%).

The great majority of rare haplotypes were detected just once in our study (HF 0.12%, 471 haplotypes), representing approx. 81.8% of all detected haplotypes.

Concluding the haplotype frequency analyses, Figure 4 presents a summary of the main HLA-A/HLA-B/HLA-C/HLA-DRB1 haplotypes across the 6-digit, 4-digit, and 2-digit resolutions (as described above).

## 3. Discussions

This study provides an extensive analysis of the HLA-A/HLA-B/HLA-C/HLA-DRB1 haplotypes within a cohort of Romanian stem cell donors, offering valuable insights into the genetic diversity and haplotype frequency distribution in this population. By employing high-resolution typing (6-, 4-, and 2-digit levels), we identified a total of 635 unique HLA-A/HLA-B/HLA-C/HLA-DRB1 haplotypes, highlighting significant heterogeneity in the dataset. The two most frequent haplotypes at the 6-digit level, HLA-A*01:01:01/HLA-B*08:01:01/HLA-C*07:01:01/HLA-DRB1*03:01:01 (HF 3.33%) and HLA-A*02:01:01/HLA-B*18:01:01/ HLA-C*17:01:01/HLA-DRB1*11:04:01 (HF 2.84%), were observed in 27 and 23 individuals, respectively. These haplotypes align with established reports of common European haplotypes, particularly those frequently associated with Caucasian populations [5,8,10,13,14,15,24,25,26,27,28]. Their presence as the top-ranking haplotypes in our cohort reinforces the notion of genetic continuity within this population segment. However, it is noteworthy that even these top-ranked haplotypes have a prevalence of only ~3%, which classifies them as low-frequency allelic combinations on a broader population scale. In fact, no haplotype exceeded 5% frequency in our sample, indicating that unlike some Western European populations where a single haplotype can reach 10% or higher, the Romanian population does not appear to have a “modal” HLA haplotype that is common in more than a small minority of individuals.

The frequency patterns were consistent across resolution levels, with only minor variations. At the 4-digit resolution, the same two haplotypes remained the most frequent, and their frequencies (3.33% and 2.84%) were essentially unchanged from the 6-digit values (since splitting into allelic subtypes did not affect those particular haplotypes). At the 2-digit resolution, however, both haplotypes exhibited a slight increase in observed frequency (to 3.46% and 2.96%, respectively), reflecting the effect of consolidating rare allele variants under broader allele groups. In other words, only when reducing to 2-digits did these haplotypes gain additional occurrences (from individuals carrying allelic variants of the same two-field groups), whereas the 4-digit resolution frequencies remained the same as the fully resolved ones.

An important observation is the highly disproportionate distribution of haplotype frequencies. Apart from the two most common haplotypes, the remaining ~99.7% of detected haplotypes each had a frequency below 1%, with the overwhelming majority categorized as rare (HF < 0.5%). In fact, the largest category consisted of haplotypes observed in a single individual (556 out of 635 haplotypes, ~87.6% of all unique 6-digit haplotypes, each with HF = 0.12%). This finding underscores the extensive genetic variability within the Romanian donor cohort. It mirrors patterns reported in other genetically diverse populations, where a plethora of unique haplotype combinations make up the genetic landscape. The predominance of single-occurrence haplotypes poses challenges for donor registries: patients with such rare haplotypes may have difficulty finding matched donors, emphasizing the need to recruit a very large and genetically varied donor pool.

The identification of two relatively frequent haplotypes (~3% each) in our study provides a point of comparison with other populations. These two haplotypes correspond to well-known extended HLA haplotypes in Europe. The HLA-A*01:01/HLA-B*08:01/HLA-C*07:01/HLA-DRB1*03:01 haplotype, for example, is a hallmark of Northwest European populations—it ranks first in frequency in countries like Ireland, the UK, and parts of Western Europe, often with frequencies around 5–10% in those populations. In our Romanian cohort, we observed this haplotype at 3.33%, which is consistent with the trend of lower frequencies toward the southeast of Europe (e.g., ~2–4% in Greece, Albania, and Macedonia) [5,8,10,13,14,15,24,25,26,27,28]. This decreasing west-to-east gradient has been attributed to historical migration patterns: the HLA-A*01:01/HLA-B*08:01/HLA-DRB1*03:01 haplotype likely spread in Europe through the migrations of Celtic and Germanic groups, and it never became as predominant in Eastern Europe as in the West.

On the other hand, the HLA-A*02:01/HLA-B*18:01/HLA-C*07:01/HLA-DRB1*11:04 haplotype (the second most frequent in our data) shows a different geographical pattern. This haplotype has its highest frequencies in the Balkans and parts of the Eastern Mediterranean [24,25,26,27,28]. Studies report it as the top-ranking haplotype in certain Balkan populations, for instance ~5–7% in Albania and specific cohorts in North Macedonia. In Western and Northern Europe, however, it is far less common (often <1%) [24,25,26,27,28]. Our finding of 2.84% in Romania aligns with Romania’s position as a geographically intermediate region—higher than in Western Europe, but lower than in the core Balkan peninsula. The prominence of this haplotype in Southeastern Europe has been linked to the genetic legacy of ancient Thracian/Illyrian populations and subsequent regional isolation [5,8,10,13,14,15].

It is also illuminating to compare these two haplotypes within the context of population substructure. In the general Macedonian population (bone marrow registry data), both haplotypes appear at a few percent each. However, in an isolated subgroup—Macedonian Muslims—one study reported the HLA-A*02:01/HLA-B*18:01/HLA-DRB1*11:04 haplotype at an exceptionally high ~7% frequency, while the HLA-A*01:01/HLA-B*08:01/HLA-DRB1*03:01 haplotype was virtually absent. This striking difference within a single country underscores how genetic drift, founder effects, and endogamy can dramatically skew haplotype distributions in isolated communities. In our Romanian sample, which is drawn from a nationwide registry and is thus more mixed, we did not observe such extreme discrepancies between subgroups. Nonetheless, these comparisons highlight that population structure can influence haplotype frequency profiles.

HLA-A*01:01:01/HLA-B*08:01:01/HLA-C*07:01:01/HLA-DRB1*03:01:01 is among the most common extended HLA haplotypes in Europe, peaking at ~14–15% frequency in Ireland (the highest observed worldwide, ranked first nationally) and remaining the top haplotype in the United Kingdom (~9–10%) as well as across much of Northern and Central Europe (e.g., ~11.5% in Sweden, ~8–12% in several Central/Western European populations, all ranked first). In contrast, its frequency declines in Southern and Eastern Europe (generally only ~2–5%), and it is no longer predominant in those regions (for instance, HF~2.8% in Spain, where it ranks fourth, and ~4.3% in Ukraine, ranking third) [10,13,24,25,26,27,28]. Historically and genetically, this haplotype—often termed the “8.1 ancestral haplotype”—holds significance as an ancient lineage that likely originated in northwestern Europe. Its present-day distribution (high in Celtic and Germanic populations of Northwestern Europe, but lower in the South and East) is thought to reflect ancient population dynamics, including Celtic expansions and later Germanic migrations. Notably, the A1-B8-DR3 haplotype has persisted far longer and at higher frequencies than would be expected under neutral evolution, spanning a large MHC segment in strong linkage disequilibrium; this suggests it has been maintained by positive selection in European populations over millennia [10,13,24,25,26,27,28].

The extended haplotype HLA-A*02:01:01/HLA-B*18:01:01/HLA-C*07:01:01/HLA-DRB1*11:04:01 shows a strikingly uneven distribution across Europe, reaching its highest frequencies in the Balkans and Mediterranean region. It is the single most common HLA haplotype in several Southeast European populations—for example, it ranks first in Macedonian Muslims (6.8% frequency), Albanians (≈5–5.6% in Albania), Kosovo Albanians (5.2%), and Bulgarians (3.6%)—and it appears as the second-most frequent haplotype in Macedonia, Greece, Croatia, and Romania (on the order of 2–3% in those populations). Outside of the Balkan/Black Sea area, its prevalence drops off sharply: in Western and Northern Europe it is exceedingly rare (on the order of 0.3–0.4% in France and Spain, and <1% in countries like Norway or Ireland), and it remains low in Eastern Europe and Russia as well (generally <1%, e.g., 0.8–1.5% in surveyed groups) [5,8,10,13,14,15,24,25,26,27,28]. This marked geographic clustering has been linked to deep historical roots in the indigenous Paleo-Balkan peoples—Thracians, Dacians, Illyrians, and Pelasgians—and to long-term genetic continuity of Balkan populations [5,8,10,13,14,15]. By contrast, Northwestern Europeans of Germanic descent (and their diaspora) more commonly carry the classic “8.1” ancestral haplotype HLA-A*01:01:01/HLA-B*08:01:01 (A1-B8-DR3), difference observed in the HLA profiles of the Transylvanian Saxon (Sași) minority and the Romanian diaspora in Germany [26,27,28], which reflect their Germanic origins (high frequencies of the A1-B8-DR3 haplotype, HF approx. 6%) with only minimal representation of the A2-B18-DR11 Balkan haplotype (HF ~2%)—consistent with medieval Germanic migrations into Romania and the more recent emigration of ethnic Romanians to Western Europe and explaining the differences in haplotypes frequencies when compared with the results from the current study.

Beyond specific haplotypes, our data indicate that Romania’s HLA profile is one of high diversity, with no single overwhelmingly common haplotype. This is similar to findings in other large, outbred populations such as those in Central and Eastern Europe. For instance, similarly to the current results, a comprehensive study in Transylvania (northwestern Romania) found that no HLA haplotype exceeded 5% frequency in that region’s donor pool [18]. Studies in Poland and Albania likewise report a long tail of low-frequency haplotypes rather than a short list of dominant ones [26,27,28]. These observations suggest that historical migrations and admixture in Eastern Europe, coupled perhaps with a lack of strong bottlenecks in recent history, have resulted in a very heterogeneous HLA makeup. This contrasts with populations that underwent founder events or long-term isolation (e.g., certain island or tribal populations), where one or a few haplotypes can reach very high frequencies due to genetic drift.

From a clinical perspective, the absence of any high-frequency haplotypes in the Romanian population has practical implications. It means that patients from this population will often have HLA combinations that are relatively rare, which could make finding perfectly matched unrelated donors more challenging on average. The fact that ~98–99% of haplotypes are under 1% frequency implies that nearly every patient may present a unique combination. This underlines the importance of expanding donor registries with as many contributors as possible from this and similar populations to improve the chances that even rare haplotype combinations are represented. It also supports efforts to recruit donors from diverse ethnic backgrounds, as genetic matches are more likely within the same ethnic or regional group [8,10,13,15].

It is important to note some limitations of our study. First, the sample size (405 individuals) provides a good initial survey but is still modest; some haplotype frequency estimates might change with a larger cohort, and very rare haplotypes could have been missed. Second, although all donors were assumed unrelated, we did not perform genetic analyses to confirm the absence of close kinship among participants. If a few related individuals were inadvertently included, certain haplotypes could appear slightly inflated in frequency. We acknowledge this and suggest that future studies perform identity-by-descent analysis to eliminate any such bias (e.g., using genome-wide SNP data or algorithms as described in population genetic frameworks). Third, our cohort was not stratified by regional ancestry within Romania; subtle population substructure (for example, differences between historical regions or ethnic subgroups) could exist. We did not conduct an ancestry-informative marker analysis or principal component analysis which could detect such substructure. Investigating this would be valuable in future research to ensure that the sample is representative of a homogeneous population or to understand differences if it is not.

Despite these caveats, our findings are robust in demonstrating that the Romanian population’s HLA haplotype distribution is extremely broad and lacks pronounced modal haplotypes. This picture is consistent with patterns observed in other European populations of similar size and diversity. The data contribute to the growing body of knowledge indicating that, for many populations, one must expect to encounter a vast array of HLA combinations. For transplantation medicine, this reinforces the need for international collaboration and donor sharing, since a Romanian patient’s best match may not reside in Romania and vice versa. It also highlights the value of high-resolution HLA typing (using NGS) in precisely identifying these haplotypes—information that can improve algorithms for virtual donor-patient matching and help predict the likelihood of finding a match in a registry.

## 4. Material and Methods

Study Population: The current study included 405 Romanian (Caucasian) voluntary donors (61% male, with an average age of 43.3 ± 7.7 years) registered in the National Registry of Voluntary Hematopoietic Stem Cell Donors (RNDVCSH) for stem cell donation between 2020 and 2021. All individuals were healthy and unrelated. Written informed consent was obtained from all subjects. The study was conducted in accordance with the Declaration of Helsinki and was approved by the Ethics Committee of Fundeni Clinical Institute (protocol code 7916, approved on 10 February 2021).

HLA Typing: Genomic DNA was extracted from EDTA-anticoagulated peripheral blood using a commercial kit (Qiagen, Hilden, Germany). Four-field high-resolution HLA genotyping for HLA-A, HLA-B, HLA-C (class I) and HLA-DRB1 (class II) loci was performed using next-generation sequencing (NGS) on the Illumina MiSeq platform (Illumina Inc., San Diego, CA, USA). Library preparation and amplification were carried out with the Mia Fora Flex 5™ NGS kit (Immucor, Warren, NJ, USA) according to the manufacturer’s instructions. Sequencing data were analyzed with the Mia Fora NGS software v4.0 (Immucor), which uses an algorithm to phase sequence reads and assigns HLA alleles based on the IPD-IMGT/HLA Database.

Haplotype Frequency Estimation: HLA-A~B~C~DRB1 haplotypes were inferred from the phaseresolved genotype data. Haplotype frequencies (HF) were estimated using an expectation-maximization (EM) algorithm embedded in the Mia Fora software, which accounts for ambiguous phase combinations. The default convergence criteria have been used and the consistency verified by repeating the EM analysis multiple times. While open-source programs such as Arlequin 3.5 are available for haplotype analysis, validated commercial software has been chosen for its specialization in HLA typing. All individuals were considered unrelated in the frequency calculation. The haplotype frequency results were expressed as a percentage of chromosomes carrying that haplotype in the population (total chromosomes = 810 for each locus).

Population Comparisons: For comparative analysis, HLA haplotype frequency data for other populations were obtained from the Allele Frequencies Net Database (AFND). We specifically drew on available European dataset entries and the relevant literature for haplotype frequencies in neighboring countries.

Statistical comparisons of allele or haplotype frequencies between groups (where applicable) were performed using Fisher’s exact test, with a significance threshold of *p* < 0.05. However, the primary focus was descriptive frequency analysis; thus, formal statistical tests were limited to key observations.

## 5. Conclusions

In conclusion, this study provides a comprehensive high-resolution analysis of HLA-A, HLA-B, HLA-C, and HLA-DRB1 haplotypes in a sample of Romanian stem cell donors. We found an exceptionally high level of haplotypic diversity, with 635 distinct haplotypes identified and no single haplotype exceeding ~3% frequency in the cohort. The two most frequent haplotypes (HLA-A01:01:01/HLA-B*08:01:01/HLA-C*07:01:01/HLA-DRB1*03:01:01 at 3.33%, and HLA-A*02:01:01/HLA-B*18:01:01/HLA-C*17:01:01/HLA-DRB1*11:04:01 at 2.84%) correspond to well-known European haplotypes, but their moderate frequencies underscore that even the “common” haplotypes in this population are relatively uncommon in absolute terms. This reflects the genetic mosaic of the Romanian population shaped by historical migrations and admixture.

Our findings carry important implications for donor registry strategies and transplant compatibility. The predominance of very low-frequency haplotypes means that patients from this population may often possess rare HLA profiles. To improve the chances of finding matched donors, it is crucial to continue expanding the diversity of the donor pool, both within national registries and through international cooperation. The use of high-resolution NGS typing, as demonstrated in our study, is valuable for uncovering the full extent of HLA diversity and should be integrated into routine donor typing to enhance match accuracy. Finally, while this study establishes a baseline for Romanian HLA haplotype frequencies, further research involving larger cohorts and multiple centers is warranted to validate and refine these frequency estimates. Such studies could also incorporate analyses of population substructure and relatedness to address the limitations noted. Overall, our work contributes to a deeper understanding of HLA population genetics in Eastern Europe and supports ongoing efforts to optimize donor matching in hematopoietic stem cell transplantation.

## Figures and Tables

**Figure 1 ijms-26-05250-f001:**
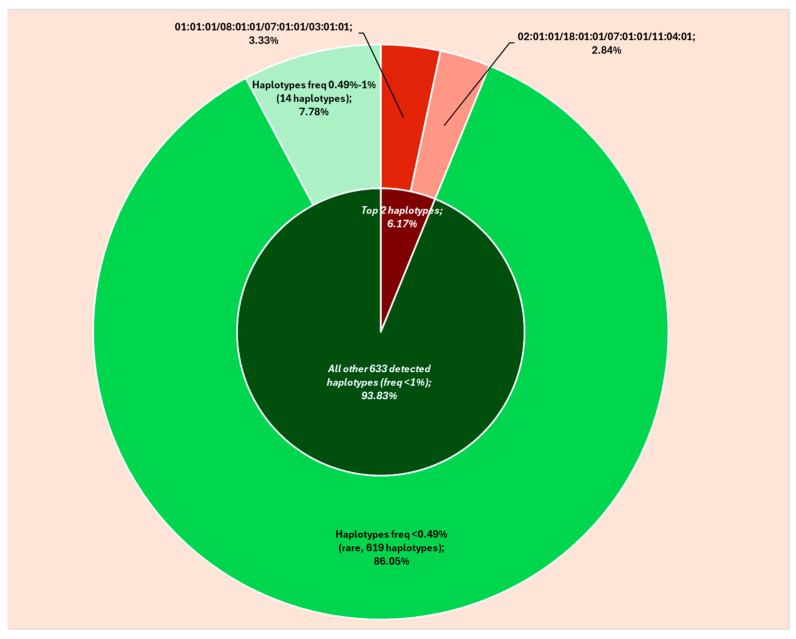
Detected HLA-A/HLA-B/HLA-C/HLA-DRB1 haplotypes (6-digits).

**Figure 2 ijms-26-05250-f002:**
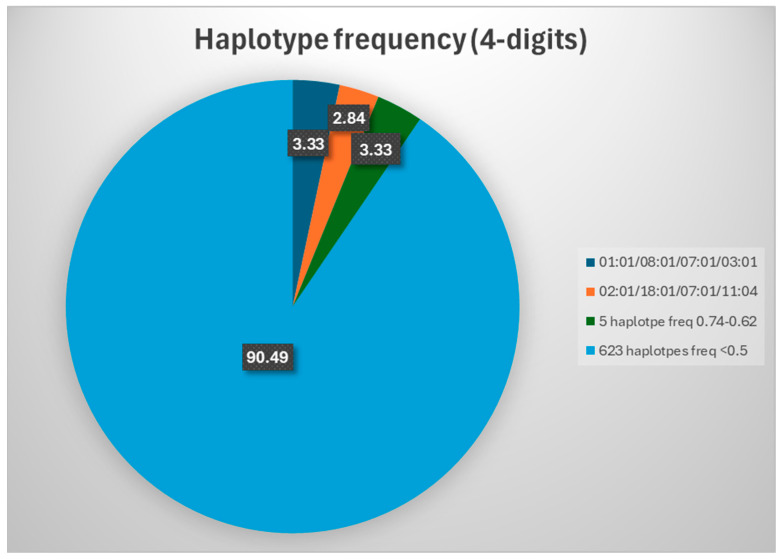
4-digit haplotype frequencies.

**Figure 3 ijms-26-05250-f003:**
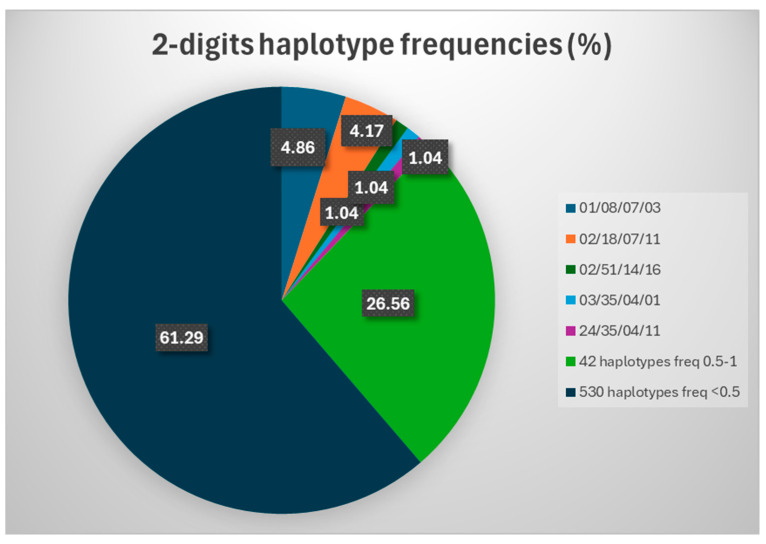
2-digit HLA-A/HLA-B/HLA-C/HLA-DRB1 haplotypes in the analyzed cohort.

**Figure 4 ijms-26-05250-f004:**
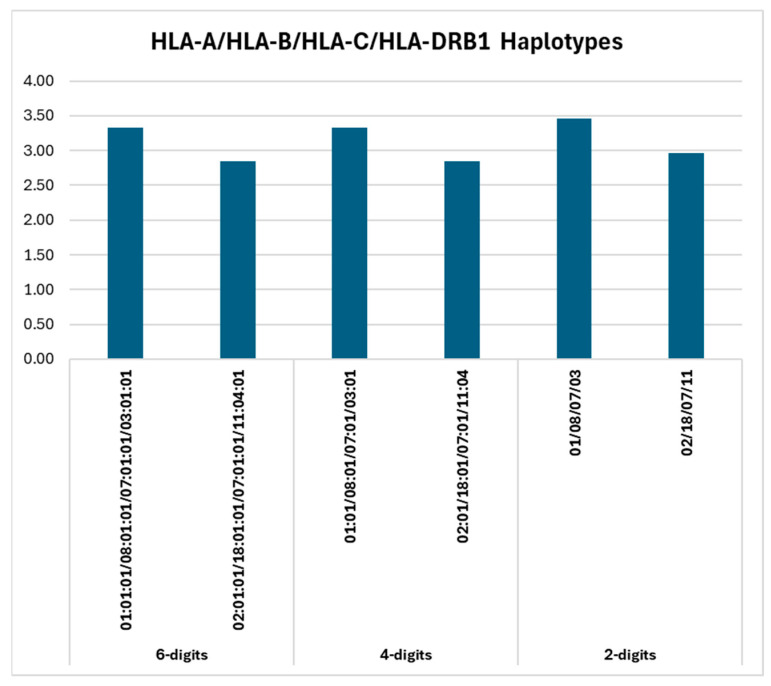
HLA-A/HLA-B/HLA-C/HLA-DRB1 haplotypes (6-, 4- and 2-digits).

**Table 1 ijms-26-05250-t001:** Top HLA-A, HLA-B, HLA-C, HLA-DRB1 alleles.

HLA-A Allele	AF (Allele Frequency)	HLA-B Allele	AF	HLA-C Allele	AF	HLA-DRB1 Alleles	AF
A*02:01:01	26.11%	B*18:01:01	11.25%	C*07:01:01	17.36%	DRB1*16:01:01	12.59%
A*01:01:01	12.50%	B*51:01:01	10.83%	C*04:01:01	13.47%	DRB1*11:04:01	12.10%
A*24:02:01	11.67%	B*08:01:01	7.78%	C*12:03:01	10.69%	DRB1*03:01:01	11.98%
A*03:01:01	9.72%	B*35:01:01	5%	C*06:02:01	9.44%	DRB1*07:01:01	9.26%
A*11:01:01, A*32:01:01	6.25%	B*35:03:01	4.58%	C*02:02:02	8.75%	DRB1*01:01:01	7.28%

**Table 2 ijms-26-05250-t002:** Top HLA-A/HLA-B/HLA-C/HLA-DRB1 haplotypes (6-digits).

HLA-A/HLA-B/HLA-C/HLA-DRB1 Haplotypes	No.	Haplotype Frequency (HF) (%)
01:01:01/08:01:01/07:01:01/03:01:01	27	3.33
02:01:01/18:01:01/07:01:01/11:04:01	23	2.84
02:01:01/51:01:01/14:02:01/16:01:01	6	0.74
24:02:01/35:02:01/04:01:01/11:04:01	6	0.74
02:01:01/27:02:01/02:02:02/16:01:01	5	0.62
03:01:01/35:01:01/04:01:01/01:01:01	5	0.62
26:01:01/38:01:01/12:03:01/13:01:01	5	0.62
01:01:01/13:02:01/06:02:01/07:01:01	4	0.49
01:01:01/52:01:01/12:02:02/15:02:01	4	0.49
01:01:01/57:01:01/06:02:01/07:01:01	4	0.49
02:01:01/07:02:01/07:02:01/15:01:01	4	0.49
02:01:01/13:02:01/06:02:01/07:01:01	4	0.49
02:01:01/44:27:01/07:04:01/16:01:01	4	0.49
23:01:01/44:03:01/04:01:01/07:01:01	4	0.49
24:02:01/13:02:01/06:02:01/07:01:01	4	0.49
25:01:01/18:01:01/12:03:01/15:01:01	4	0.49

**Table 3 ijms-26-05250-t003:** Top HLA-A/HLA-B/HLA-C/HLA-DRB1 haplotypes (4-digits).

HLA-A/HLA-B/HLA-C/HLA-DRB1 Haplotypes	No.	Haplotype Frequency (%)
01:01/08:01/07:01/03:01	27	3.33
02:01/18:01/07:01/11:04	23	2.84
02:01/51:01/14:02/16:01	6	0.74
24:02/35:02/04:01/11:04	6	0.74
02:01/27:02/02:02/16:01	5	0.62
03:01/35:01/04:01/01:01	5	0.62
26:01/38:01/12:03/13:01	5	0.62
01:01/13:02/06:02/07:01	4	0.49
01:01/52:01/12:02/15:02	4	0.49
01:01/57:01/06:02/07:01	4	0.49
02:01/07:02/07:02/15:01	4	0.49
02:01/13:02/06:02/07:01	4	0.49
02:01/44:27/07:04/16:01	4	0.49
11:01/55:01/03:03/16:01	4	0.49
23:01/44:03/04:01/07:01	4	0.49
24:02/13:02/06:02/07:01	4	0.49
25:01/18:01/12:03/15:01	4	0.49

**Table 4 ijms-26-05250-t004:** Top HLA-A/HLA-B/HLA-C/HLA-DRB1 haplotypes (2-digits).

HLA-A/HLA-B/HLA-C/HLA-DRB1 Haplotypes	No.	Haplotype Frequency (%)
01/08/07/03	28	3.46
02/18/07/11	24	2.96
02/51/14/16	6	0.74
03/35/04/01	6	0.74
24/35/04/11	6	0.74
02/08/07/03	5	0.62
02/13/06/07	5	0.62
02/27/02/16	5	0.62
02/35/04/11	5	0.62
02/35/04/14	5	0.62
02/44/07/16	5	0.62
26/38/12/13	5	0.62
32/35/04/11	5	0.62
01/13/06/07	4	0.49
01/52/12/15	4	0.49
01/57/06/07	4	0.49
02/07/07/15	4	0.49
02/52/12/15	4	0.49
11/35/04/11	4	0.49
11/55/03/16	4	0.49
23/44/04/07	4	0.49
24/13/06/07	4	0.49
24/35/04/16	4	0.49
25/18/12/15	4	0.49

## Data Availability

Data is contained within the article.

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
