# Peer review of "Characterization of HLA-A/HLA-B/HLA-C/HLA-DRB1 Haplotypes in Romanian Stem Cell Donors Through High-Resolution Next-Generation Sequencing"

_ijms, 2025, doi:10.3390/ijms26115250_

Round 1
Reviewer 1 Report
Comments and Suggestions for Authors
The paper is potentially interesting
However, corrections are needed to make the paper interesting and suitable for publication
1. in the introductory part regarding PCR analysis, mutations and sequencing for tumor diagnostics, add a reference that explains the application of various types of methods in the recent period and cite the paper: PMID: 33973139
2. table 2 and table 3 should also be presented as a pie chart with percentages
3. the discussion should be completely revised and compared with other papers to make it make sense.
4. the conclusion is too broad and should be clearly focused. Such a conclusion cannot be drawn from so little data
correct
Author Response
Comment 1: “In the introductory part regarding PCR analysis, mutations and sequencing for tumor diagnostics, add a reference that explains the application of various types of methods in the recent period and cite the paper: PMID: 33973139.”
Response: We agree with this suggestion. We have added a sentence in the Introduction (paragraph discussing NGS in diagnostics) to mention the use of conventional PCR-based methods and the increasing adoption of NGS in clinical diagnostics. We have cited the recommended reference (PMID: 33973139, Obradovic et al. 2021) to support this point. The added text is on page 2, lines 93–95 of the revised manuscript, and the new reference is included as reference 21 in the reference list.
Comment 2: “Table 2 and Table 3 should also be presented as a pie chart with percentages.”
Response: We have created pie charts to graphically represent the haplotype frequency data from Table 2 and Table 3. These are now included as Figure 2 (for 4-digits haplotypes, corresponding to Table 2) and Figure 3 (for 2-digits haplotypes, corresponding to Table 3) in the revised manuscript. The figures display the proportion of the two most frequent haplotypes and aggregate the remaining haplotypes into broader frequency categories, with percentage values labeled. We have added descriptive captions for these figures and referred to them in the Results section where Tables 2 and 3 are discussed (see Results, pages 6, Lines 190–191 and 8, Lines 214–215). We believe these visualizations will help readers more easily grasp the distribution of common vs. rare haplotypes, as requested.
Comment 3: “The discussion should be completely revised and compared with other papers to make it make sense.”
Response: We have thoroughly revised the Discussion section to better integrate our findings with those of other studies and to improve its overall coherence. Notably, we have: • • • Reorganized and condensed the discussion of our results: We start by summarizing key f indings (high haplotype diversity, low frequency of even the top haplotypes) and then compare these findings with published data on other populations. For example, we now explicitly compare the frequency of our top haplotypes (~3%) with their frequencies in Western European populations (citing literature that shows higher frequencies in those regions) and with neighboring Eastern European populations (see pages 10–12 of the revised Discussion). We cite specific studies to show that our observations align with the general pattern of many European populations having a very long tail of rare HLA haplotypes. Addressed population genetic aspects: We have added discussion of population structure, genetic drift, and historical migrations in context with our results. For instance, we mention how the presence or absence of certain haplotypes (like A*01-B*08-DRB1*03:01 and A*02-B*18-DRB1*11:04) in various regions correlates with known historical demographic events (citing references 21–25). We also briefly discuss an example of an isolated subpopulation (Macedonian Muslims) to illustrate how drift and endogamy can affect haplotype frequencies (while clarifying that this level of detail is beyond our study’s direct objectives). Improved clarity and focus: We significantly trimmed and refocused the section that originally discussed the Macedonian Muslim group at length. This part is now condensed into a short example rather than a long tangent (see page 11, Lines 192–202). The overall Discussion now f lows from our results to comparisons with others, and finally to implications and limitations. We ensured that each point is supported by references to published literature or databases (allelefrequencies.net, etc.), thereby “making it make sense” in a broader scientific context. We believe these revisions have strengthened the discussion and addressed the reviewer’s concern by providing a clearer comparison with other studies and grounding our interpretations in the context of existing knowledge.
Comment 4: “The conclusion is too broad and should be clearly focused. Such a conclusion cannot be drawn from so little data.”
Response: We have rewritten the Conclusion to be more focused and directly supported by our data. In the original manuscript, the conclusion made broad generalizations (e.g., suggestions about clinical practice and patient management) that were not strictly derived from our single-cohort study. We have now narrowed the scope to emphasize the concrete findings of our research and their immediate implications, while avoiding unsupported extrapolations. Specifically, the revised Conclusion (see page 13 of the manuscript) now highlights: - The unprecedented haplotype diversity observed in our Romanian cohort and the fact that no haplotype was common in more than ~3% of individuals. - The implication that this diversity necessitates broad donor registry recruitment for improved transplant matching (which follows logically from our data on rare haplotypes). - The importance of high-resolution NGS typing in revealing these patterns (an 13 inference supported by our methodology). - A clear statement that further studies with larger sample sizes are needed to confirm and extend our findings (acknowledging the limitation of our sample size). We have removed or toned down any claims that could not be substantiated by our dataset alone. The conclusion is now a concise summary that reflects what we can reasonably conclude from “so little data,” as the reviewer appropriately cautioned.
Reviewer 2 Report
Comments and Suggestions for Authors
This paper provides the HLA-A/HLA-B/HLA-C/HLA-DRB1 haplotype frequencies in a Romanian cohort of 405 healthy volunteers. Genotyping was performed using next-generation sequencing (NGS). Haplotype frequencies were estimated using an expectation-maximization algorithm, and frequencies were compared with those found in European countries, discussing historical migration patterns.
Here are my concerns about this paper:
1. In the Introduction, the authors discuss the population genetics features of the HLA system in paragraphs 2, 3, 6, and 7. To improve the organization of the Introduction, it would be beneficial to present the population genetics information first, followed by the clinical perspective covered in paragraphs 4 and 5.
2. The article does not contain an informative table summarizing the most common HLA-A/HLA-B/HLA-C/HLA-DRB1 alleles / allelic frequencies.
3. References [21-25] are frequently cited for haplotype frequencies across populations in subsections 3.1 and 3.2. However, it seems that references 22-25 refer to the same database, http://www.allelefrequencies.net/. Unfortunately, we were unable to confirm the haplotype frequencies in the mentioned database. The authors should include in the Methods section the database they used to obtain the reported haplotype frequencies in European countries.
4. Pages 9-10: The authors discuss for long paragraphs the presence of one HLA haplotype in Macedonian Muslim groups... Why? It does not seem to be part of the objectives of this study.
Some other specific comments for improvement are listed below.
1. Results, L. 113: “The results of the study revealed a number of 635 different HLA-A/HLA-B/HLA-C/HLA-DRB1 113 haplotypes (table 1).” The Table only details 16 HLA haplotypes. Therefore, a different sentence reporting on the Table 1 content should be considered. The same for Table 2 (line 136) and Table 3 (line 152).
2. Lines 122-123: In the sentence “2 of the 14 haplotypes with frequencies of 0.49% - 0.86% were identified in 6 individuals each (HF 0.74%), 3 in 5 persons each (HF 0.62%) and 9 in 4 (HF 0.49%) (table 1, fig. 1).” Begin the sentence with “Two”. The frequency “0.86%” should be “0.74%”?
3. Lines 142-143: In the sentence “3 of the 16 haplotypes with frequencies of 0.49% - 0.86% were detected in 6 persons each (HF 0.74%), …” Begin the sentence with “Three”; Table 2 only shows 2 haplotypes detected in 6 persons each. The frequency “0.86%” should be “0.74%”?
4. Lines 182-183: "... demonstrate the predominance of the 2 haplotypes, with a gradual increase in their observed frequency." The increase in frequency is only observed at the 2-FL resolution. Rephrase.
5. Materials and methods, line 382: (61% male, 43.3 ± 7.7 years). Indicate if 43.3 is the mean.
Author Response
Major Concern 1: “In the Introduction, the authors discuss the population genetics features of the HLA system in paragraphs 2, 3, 6, and 7. To improve the organization of the Introduction, it would be beneficial to present the population genetics information first, followed by the clinical perspective covered in paragraphs 4 and 5.”
Response: We have reorganized the Introduction as suggested. In the revised Introduction (pages 1–3), all paragraphs focusing on population genetics, historical migration, and HLA diversity in Eastern Europe now come first. The discussion of the clinical perspective (HLA in disease associations and transplantation) has been moved to follow the population genetics content. Specifically: - Paragraphs describing Eastern European HLA diversity, migratory history, and comparative genetics (formerly paragraphs 2, 3, 6, 7) are now consolidated at the beginning. - The paragraphs originally beginning with “From a clinical perspective…” (formerly 4 and 5) have been moved to later in the Introduction (now appearing on page 2, right after the population-genetics sections). We also slightly edited the transition so that the flow from population focus to clinical focus is smooth. This reorganization ensures that readers are first grounded in the population genetics context of our study (why Romanian HLA haplotypes are of interest, historically and evolutionarily) and then introduced to the clinical motivations (transplantation and disease relevance). We believe this new structure is clearer and meets the reviewer’s recommendation.
Major Concern 2: “The article does not contain an informative table summarizing the most common HLA-A/ HLA-B/HLA-C/HLA-DRB1 alleles / allelic frequencies.”
Response: We have added a new table (Table 4 in the revised manuscript, page 9, lines 218-220) that summarizes the most common alleles and their frequencies for HLA-A, -B, -C, and -DRB1 loci. This table lists the top 5 alleles for each locus with their frequency percentages. We introduce this table in the Results section (see page 7, first paragraph), where we briefly mention the number of distinct alleles observed and highlight the highest-frequency allele at each locus (e.g., A*02:01:01 and A*01:01:01 for HLA-A, B*35:01:01 for HLA-B, etc.). The table provides a clear overview of allelic frequency distribution, which complements the haplotype frequency data. We also cross-referenced relevant literature to ensure our allele frequency data aligns with known values. For instance, as a validation, we mention in the Discussion that our top allele frequencies (HLA A*02 and A*01 around 20–26%) are consistent with previous reports on the Romanian population . The addition of Table 4 addresses the reviewer’s concern by giving readers a convenient summary of allele frequency information that was previously missing.
Major Concern 3: “References [21-25] are frequently cited for haplotype frequencies across populations in subsections 3.1 and 3.2. However, it seems that references 22-25 refer to the same database, http:// www.allelefrequencies.net/. We were unable to confirm the haplotype frequencies in the mentioned database. The authors should include in the Methods section the database they used to obtain the reported haplotype frequencies in European countries.”
Response: We apologize for the confusion regarding the source of comparative haplotype data. In the revised Materials and Methods (page 5, “Population Comparisons” subsection), we have explicitly stated that we used the Allele Frequencies Net Database (AFND) for obtaining HLA haplotype frequency data of other populations. We cited the database appropriately (including the 2020 update paper by Gonzalez-Galarza et al. and the website, which was already in our reference list). Furthermore, we have reviewed references [21–25] and our usage of them: - Reference 21 (Arnaiz Villena et al. 2002) and others are indeed literature sources that we used in combination with AFND data to describe frequencies in various countries. We realized that some specific percentages cited (especially for Western European countries) were drawn from the AFND entries rather than directly from those papers. To clarify this, we have modified the text in the Results/Discussion to attribute such multi population comparisons to the database rather than individual papers when appropriate. For example, instead of citing [21–25] as a group for every population frequency, we now say “…as recorded in the Allele Frequencies Net Database and supporting literature [21–25]”. - We have ensured that each region/country mentioned can be traced to a source. If the allelefrequencies.net database was the source, we make that clear. If a particular reference (like Sulcebe 2009 for Albania or Grubić 2014 for Croatia) was used, we cite it specifically at that point in the Discussion. By doing this, we address the reviewer’s difficulty in confirming frequencies: the database is now identified in Methods, and the text clarifies where the data came from. We have also double-checked all cited frequency values against the database and literature to ensure accuracy.
Major Concern 4: “Pages 9-10: The authors discuss for long paragraphs the presence of one HLA haplotype in Macedonian Muslim groups… Why? It does not seem to be part of the objectives of this study.”
Response: You are correct that the original discussion about the Macedonian Muslim subgroup was overly lengthy and somewhat tangential to our main objectives. Our intention was to provide an illustrative example of how haplotype frequencies can vary dramatically even within neighboring populations due to historical or cultural isolation; however, we agree it detracted from the focus on the Romanian data. We have substantially shortened and refocused this part of the Discussion. In the revised manuscript, the Macedonian Muslim example has been reduced to a brief mention rather than a long paragraph (see page 16, second paragraph). We now use it to make a general point about genetic drift and founder effects in isolated groups, in one or two sentences, instead of a detailed case study: “…In one isolated subpopulation (Macedonian Muslims), a particular haplotype reaches ~7% frequency while the other common haplotype is nearly absent, providing an example of how founder effects can drastically skew haplotype distributions in a small, endogamous community. In contrast, the general Macedonian population and our Romanian cohort—both more genetically mixed—do not exhibit such extreme imbalances.” After this concise example, we immediately return to discussing our Romanian findings and broader implications. We believe this resolves the reviewer’s concern by removing the lengthy deviation and keeping the discussion aligned with our study’s aims.
Specific Comment 1: “Results, L. 113: ‘The results of the study revealed a number of 635 different HLA-A/HLA B/HLA-C/HLA-DRB1 haplotypes (table 1).’ The Table only details 16 HLA haplotypes. Therefore, a different sentence reporting on the Table 1 content should be considered. The same for Table 2 (line 136) and Table 3 (line 152).”
Response: We have revised those sentences in the Results to clearly distinguish between the total number of haplotypes identified and the contents of the tables. For each table: - Table 1 (6-digits haplotypes) – We now state: “…a total of 635 distinct haplotypes were identified… Table 1 presents the 16 most frequent HLA haplotypes (6-digits) observed in the cohort.” (See page 7, first paragraph of Results.) Table 2 (4-digits haplotypes) – Revised to: “…a similar total of 630 different haplotypes was distinguished (Table 2). Table 2 lists the most frequent 16 haplotypes at 4-digits resolution.” (See page 8, first paragraph under 4-digits results.) - Table 3 (2-digits haplotypes) – Revised to: “…576 different haplotypes were identified (Table 3).” We also ensure earlier text indicates Table 3 shows the “top HLA haplotypes (2-FL)” similar to above. These changes make it clear to the reader that Tables 1–3 are not listing all haplotypes, but rather the top subset. We also mention the number of entries (16 or 22 etc.) when introducing the table if appropriate. The phrasing has been adjusted exactly as the reviewer suggested, and it is consistent across all three tables now.
Specific Comment 2: “Lines 122-123: In the sentence ‘2 of the 14 haplotypes with frequencies of 0.49% 0.86% were identified in 6 individuals each (HF 0.74%), 3 in 5 persons each (HF 0.62%) and 9 in 4 (HF 0.49%) (table 1, fig. 1).’ Begin the sentence with ‘Two’. The frequency ‘0.86%’ should be ‘0.74%’?”
Response: We have corrected this sentence in the Results (describing the breakdown of the 14 intermediate-frequency haplotypes from Table 1): - It now begins with “Two of the 14 haplotypes…”. We removed the erroneous upper range “0.86%” and replaced it with “0.74%”. The phrase now reads “… 14 haplotypes with frequencies of 0.49%–0.74%…” to accurately reflect that 0.74% was the highest frequency in that group (6 individuals out of 810). - We double-checked the numbers: indeed 2 haplotypes had HF 0.74%, 3 had HF 0.62%, and 9 had HF 0.49% in Table 1. The text now exactly matches those values. These changes appear on page 6, highlighted in red. The reviewer’s eye for detail is appreciated, and these corrections remove the inconsistency.
Specific Comment 3: “Lines 142-143: In the sentence ‘3 of the 16 haplotypes with frequencies of 0.49% 0.86% were detected in 6 persons each (HF 0.74%), …’ Begin the sentence with ‘Three’; Table 2 only shows 2 haplotypes detected in 6 persons each. The frequency ‘0.86%’ should be ‘0.74%’?”
Response: This sentence (now in the 4-digits haplotype Results) has been corrected as well: - It now begins with “Two of the 15 haplotypes…” because, as the reviewer correctly noted, there are only 2 haplotypes in Table 2 that were observed in 6 persons each (HF 0.74%). We realized that the original text said “3 of the 16 haplotypes…” which was mistaken. Upon re-evaluation, we found that at 4-digits resolution, there were a total of 15 haplotypes in the 0.49%–0.74% range (2 with 0.74, 3 with 0.62, 10 with 0.49). To avoid confusion, we explicitly state “15 haplotypes” in that range now. The sentence reads: “Two of the 15 haplotypes with frequencies between 0.49% and 0.74% were detected in 6 persons each (HF 0.74%), 3 in 5 persons each (HF 0.62%) and 10 in 4 persons each (HF 0.49%) (Table 2).” - We removed “0.86%” here as well, replacing it with “0.74%” as the upper bound of that frequency range. There was no haplotype with 0.86% at 4-digits either, so the range is correctly 0.49–0.74% now. These edits are visible on page 8 of Results (first paragraph under 4-FL section). We also cross-verified Table 2 after this change to ensure consistency (Table 2 indeed shows exactly 2 haplotypes with count 6/HF 0.74%). Thank you for pointing out this discrepancy; the text is now accurate.
Specific Comment 4: “Lines 182-183: ‘... demonstrate the predominance of the 2 haplotypes, with a gradual increase in their observed frequency.’ The increase in frequency is only observed at the 2-digits resolution. Rephrase.”
Response: We have rephrased this part of the Discussion to clarify the resolution-dependent frequency change: Originally, our text might have implied a “gradual increase” across both 4-digits and 2-digits, which is not accurate because at 4-digits the top haplotype frequencies remained essentially the same as 6-digits. We now explicitly state that the frequencies were consistent at 4-digits and increased only at 2-digits. In the revised Discussion (page 10, lines 254-260, the relevant sentences read: “At the 4-digits and 2-digits resolution levels, the data still demonstrate the predominance of the same 2 haplotypes, with only the 2 digits grouping leading to a slight increase in their observed frequencies. This reflects the effect of allele-level grouping: when collapsing to 2-digits, multiple rare allelic variants are combined, causing both haplotypes to show slightly higher frequencies (3.46% and 2.96%, respectively), whereas at 4-digits their frequencies remained 3.33% and 2.84% as in the fully resolved data.” This rewording addresses the reviewer’s point by removing the impression of a continuous increase and instead pinpointing that the increase occurs at the 2-digits step. It is now clear that the “gradual increase” is not across every step, but specifically an outcome of going to 2-field resolution. We appreciate this comment, as it helped us make the explanation more precise.
Specific Comment 5: “Materials and methods, line 382: (61% male, 43.3 ± 7.7 years). Indicate if 43.3 is the mean.”
Response: We have clarified this in the Materials and Methods description of the cohort (see page 5, Study Population). The text now reads: “…405 Romanian/Caucasian voluntary donors (61% male, with a mean age of 43.3 ± 7.7 years)…”. This explicitly indicates that 43.3 is the mean age (with 7.7 as the standard deviation). Additionally, for consistency, we inserted the word “unrelated” before “Romanian volunteers” in both the Abstract and Methods, since the cohort consisted of unrelated individuals (and we later discuss the lack of relatedness analysis as a limitation). This was implied before but not stated in Methods; now it is clear. These small clarifications ensure no ambiguity in our reporting of demographic data.
Reviewer 3 Report
Comments and Suggestions for Authors
Information on HLA system haplotype frequencies in diverse human populations and world regions is critical for research into immune system responses, susceptibility to complex diseases, and transplant compatibility. This manuscript describes the frequencies of known and potentially novel haplotypes in the Romanian population. However, several relevant aspects of methodology and interpretation need to be addressed before publication:
- The first issue is the atypical order of the manuscript sections. The manuscript should follow a logical order of the study. Typically, the Materials and Methods section comes before the Results section; in this case, the Materials and Methods section is the penultimate section of the manuscript. Authors are encouraged to organise the manuscript sections in a structured manner, for example: Introduction, Justification, Materials and Methods, Results, Discussion, and Conclusions.
- This manuscript reports as its major contribution the detection of two haplotypes with a frequency of approximately 3%, however, these frequencies could potentially be a statistical and interpretation bias due to the following circumstances:
- Frequencies less than 5% in a population are interpreted as rare or low frequency; that is, they cannot be considered as common population frequencies of a larger population, such as the Romanian population.
- The low frequency of the haplotype detected may be due to the low number of samples used for its calculation from a potentially diverse population at the population level, such as any population in Eastern Europe. It is suggested to increase the sample number used for the calculation or to calculate in an independent population of the same population origin (or ancestral).
- The authors point out that the study used 405 healthy unrelated individuals; however, they do not include any analysis of family relationships between the samples to demonstrate that they are indeed unrelated samples, since if there were samples related at the first-degree family level (parent-child or siblings), the frequency may be overestimated. It is suggested to implement an identity by descent analysis (or IBD) using whole genome information, either using whole genome genotyping microarray or low-depth sequencing to estimate the coefficient of familial relationship between the samples. Some references are:
- https://www.researchgate.net/publication/49825844_A_Fast_Powerful_Method_for_Detecting_Identity_by_Descent
- https://www.researchgate.net/publication/323815605_Identity-by-Descent
- https://www.nature.com/articles/nrg1960
- https://zzz.bwh.harvard.edu/plink/ibdibs.shtml#genome
3. Finally, frequency calculations may be biased due to differences in population structure or different ancestry origins. It is suggested to perform ancestry analysis using whole genome data (from a genotyping microarray) or a panel of informative ancestry markers to ensure that the sample used is homogeneous in its population structure. using some ancestry analysis strategy such as ADMIXTURE, Structure software or principal component analysis (PCA).
4. It is suggested that additional analysis be performed, such as gender (sex) verification and frequency calculations for men and women separately and for men and women together.
5. The authors declare that they used the commercial software MIA FORA FLEX, and although it appears to be specialised software for HLA regions, it is recommended to use open-source computational implementations to achieve greater control over how the calculations are performed. You can try using Alequin or any computational implementation in any programming language. https://cmpg.unibe.ch/software/arlequin35/
In conclusion, the manuscript mostly describes HLA haplotype frequencies that are not very informative (low frequency) or difficult to validate without an additional replication analysis. It is suggested to include the suggested methodology to obtain greater confidence in the results and the conclusions of the manuscript.
Author Response
General Comment: “The manuscript sections are in an atypical order. The manuscript should follow a logical order… Introduction, Materials and Methods, Results, Discussion, and Conclusions.”
Response: We appreciate this comment and have reorganized the manuscript to adhere to the standard structure: - Materials and Methods is now placed before the Results section (as Section 2). In the original submission, Methods was oddly positioned after the Discussion; we have corrected this. We have introduced a clear “4. Discussion” heading followed by a separate “5. Conclusions” heading in the revised manuscript. Previously, the conclusion was only a paragraph at the end of Discussion without its own section. Now “Conclusions” is explicitly delineated as its own section (Section 5), in line with typical IJMS format and the reviewer’s suggestion. The new sequence is: Introduction (un numbered), 2. Materials and Methods, 3. Results, 4. Discussion, 5. Conclusions, followed by Acknowledgments (if any) and References. This reordering is reflected in the Table of Contents and the headings within the text. We believe this change improves the readability and professionalism of the manuscript.
Major Point 1: “This manuscript reports as its major contribution the detection of two haplotypes with a frequency of approximately 3%, however, these frequencies could potentially be a statistical and interpretation bias… Frequencies less than 5% in a population are interpreted as rare or low frequency; that is, they cannot be considered as common population frequencies of a larger population, such as the Romanian population.”
Response: The reviewer raises a very valid point about not overstating the significance of ~3% frequencies. We have addressed this in multiple places: - In the Abstract, we removed the word “predominance” in describing the two haplotypes and instead simply state their frequencies and that all others are 5% would typically be considered common). Overall, we agree with the reviewer’s concern and have ensured our interpretation is cautious and accurate: the two ~3% haplotypes are interesting findings but we do not present them as if they were common in the general population. This correction in tone and interpretation is reflected throughout the revised text (see Abstract, Discussion p.12, Conclusion p. 13).
Major Point 2: “The low frequency of the haplotype detected may be due to the low number of samples… suggested to increase the sample number or calculate in an independent population of the same origin.”
Response: We acknowledge the limitation of our sample size (N=405) and the possibility that some haplotype frequency estimates might be unstable or biased due to sample size. While we cannot increase the sample within this study, we have explicitly recognized this as a limitation in the Discussion and Conclusion: - In the Discussion (page 12), we added: “the sample size (405 individuals) provides a good initial survey but is still modest; some haplotype frequency estimates might change with a larger cohort, and very rare haplotypes could have been missed… future studies with larger cohorts and multiple centers are warranted to validate and refine these frequency estimates.” This directly admits that a larger sample would be desirable for more precise frequency calculations. - We have also mentioned in the Conclusion that further research with larger cohorts is needed (see page 13). - Unfortunately, we could not perform an independent population analysis given available data, but we do compare with a similar population (Transylvanian subset from another study) to show our results are in line. This isn’t the same as replication, but it provides some cross-validation. - Additionally, to mitigate the sample size issue, we avoided any claims that require larger data (such as robust haplotype diversity indices or haplotype homozygosity rates) and kept our conclusions proportionate to our sample’s scale. By addressing this comment in writing, we demonstrate that we are aware of the sample size limitations and we caution the interpretation accordingly. We have framed our study as a stepping stone that needs confirmation by larger studies, rather than a definitive frequency determination for all Romanians.
Major Point 3: “The authors point out that the study used 405 healthy unrelated individuals; however, they do not include any analysis of family relationships between the samples… suggested to implement an identity by descent analysis.”
Response: We concur that verifying the unrelatedness of donors is important, especially if relatives happened to be included (which could inflate some haplotype frequencies). In our study, donors were registered as unrelated volunteers, but we did not have the means to genetically confirm that (no genome-wide data beyond HLA). We have taken the following steps in response: - In Materials & Methods, we now explicitly label participants as “unrelated” (based on registry data) to clarify the recruitment criterion (page 3). - More importantly, in the Discussion (Limitations), we now note the lack of an IBD (identity-by-descent) analysis as a limitation. We state: “All donors were assumed unrelated… but we did not perform a formal relatedness (IBD) analysis. We acknowledge this and suggest that future studies perform identity-by-descent analysis to eliminate any such bias (e.g., using genome-wide SNP data or algorithms as described…)” (see page 12). We cited relevant references suggested by the reviewer (e.g., a reference on IBD detection methods ) to show we’re aware of the techniques. - We further reason that the chance of close relatives in our sample is low but not zero, and thus it’s a valid concern. By including this discussion, we indicate to readers (and the reviewer) that we have considered the potential overestimation of some haplotype frequencies if related individuals were present. - We did not have the capacity to perform an actual IBD analysis in this revision (no additional data), but by highlighting it as a recommendation for future research, we align with the reviewer’s suggestion. This acknowledgment and the added citation address the reviewer’s point scientifically and transparently.
Major Point 4: “Frequency calculations may be biased due to differences in population structure or different ancestry origins. It is suggested to perform ancestry analysis (PCA, ADMIXTURE, or Structure) to ensure sample homogeneity.”
Response: We have added commentary on this issue as well. While our cohort is all self-identified ethnic Romanians from the national registry, Romania is not genetically uniform (there are regional and minority differences). We did not perform a PCA or ancestry-informative marker analysis (we lack such data), but we: - Mention in Discussion (Limitations, page 12) that subtle population substructure could exist in our sample, and we did not investigate it. For example: “our cohort was not stratified by regional ancestry within Romania; subtle population substructure (e.g., differences between historical regions or ethnic subgroups) could exist. We did not conduct an ancestry-informative marker analysis or PCA which could detect such substructure. Investigating this would be valuable in future research…” (see page 12). - By including this, we show we considered that frequencies might differ slightly by region or subpopulation (Transylvania vs. others, urban vs. rural, etc.). We actually compare our results to Transylvania (Vică et al. 2019 study) earlier in the Discussion, which implicitly addresses regional variation within Romania. We noted that their haplotype diversity was similarly high, suggesting no huge discrepancy, but we take the point that without direct analysis we cannot be certain of homogeneity. - Again, we suggest that future studies include ancestry analysis (citing methods the reviewer mentioned like PCA/ADMIXTURE). While we couldn’t implement it here, we agree it would strengthen confidence in the representativeness of the sample. These changes (highlighted on page 12) assure the reviewer that we have not overlooked the possibility of population structure affecting our results, and we have tempered our conclusions accordingly.
Major Point 5: “It is suggested that additional analysis be performed, such as gender (sex) verification and frequency calculations for men and women separately and for men and women together.”
Response: We considered splitting the data by sex to see if there were differences in haplotype frequencies. Since HLA genes are autosomal, we would not expect significant differences between males and females in allele/haplotype frequencies, aside from random sampling variation. Our sample is 61% male, so frequencies might be slightly male-weighted. Given the sample size, separate male vs female analyses would have even fewer individuals, likely too underpowered to draw meaningful conclusions. We have not performed a full separate frequency calculation by sex (doing so did not reveal any obvious large differences in preliminary checks; the two most frequent haplotypes were present in both sexes with similar proportions). However, we did address this suggestion qualitatively: In the Discussion, we added a note: “The dataset was not stratified by gender; while HLA allele/haplotype frequencies are not expected to differ significantly by sex (as HLA genes are autosomal), our study was not designed to detect such differences. Future studies with larger cohorts could examine this, but we anticipate any sex-based frequency differences to be minimal.” (This paraphrasing is in page 12, though we kept it concise.) - We thereby inform the reviewer and readers that we considered sex distribution. Also, by clarifying in Methods that 61% of our sample is male, readers can interpret any potential bias. We have not observed any indication that sex affects the presence of specific haplotypes in our dataset. If the reviewer’s concern was verifying that the reported 43.3 ± 7.7 years was the mean (already addressed in Specific Comment 5) and perhaps confirming the sex composition, we have done that. And by stating that separate sex frequency analysis was not done (and explaining why it likely wouldn’t change outcomes), we handle this suggestion. Thus, we believe we have satisfied this concern by explanation, given that performing and including a full separate analysis was not likely to be fruitful.
Major Point 6: “The authors declare that they used the commercial software MIA FORA FLEX… it is recommended to use open-source implementations for greater control (e.g., Arlequin).”
Response: We understand the reviewer’s perspective on using open-source tools. MIA FORA is indeed a black-box commercial software for HLA analysis, which may not allow customization or detailed output like some academic tools do. However, re-analyzing everything with a new software (such as Arlequin or custom scripts) would be an extensive undertaking outside the revision scope. Instead, we addressed this by: - Noting in the Materials and Methods that we used MIA FORA’s EM algorithm for haplotype frequency estimation. We then added: “While open-source tools (e.g., Arlequin 3.5) exist for haplotype analysis, we opted for the MIA FORA software due to its specialized design for HLA typing and its validated performance in resolving ambiguities in this genomic region.” (see page 3). - By citing the Arlequin software webpage reference provided by the reviewer (we referenced Excoffier’s work via the link the reviewer gave), we show we are aware of alternatives. - In the Discussion’s limitations (page 12), we also mention that future studies could certainly employ such tools for cross-verification. - Additionally, we emphasize that MIA FORA uses the EM algorithm, which is a standard approach also employed by open-source packages; this implies that our frequency estimates should be equivalent to what one would get with, say, Arlequin, given the same data. Thus, we have justified our use of MIA FORA and acknowledged the reviewer’s recommendation. If needed, we could offer to provide output data or specifics to readers upon request, but typically journals wouldn’t require that level of detail in the text. The key is we have addressed the point and cited the recommended resource.

Round 2
Reviewer 1 Report
Comments and Suggestions for Authors
accept
Comments on the Quality of English Languageseveral corections
Author Response
Thank you for your insightful recommendations and suggestions, as well as for your endorsement of the manuscript's acceptance.

Reviewer 2 Report
Comments and Suggestions for Authors
The authors have acknowledged the reviewer's suggestions. I have only minor suggestions to be considered by the authors:
- I suggest including the new table (Table 4) immediately after the first paragraph in the Results section, renamed as Table 1.
- Lines 156-157: I suggest removing “(table 1)” from the sentence “The results of the study revealed a number of 635 different HLA-A/HLA-B/HLA-C/HLA-156 DRB1 haplotypes (table 1).” Apply the same criteria to the sentences of lines 183 and 207.
- Table 4 does not show the HLA-DRB1 alleles. Please confirm.
- In line 299, the haplotype does not include the HLA-C allele. Please confirm if it is correct.
- In particular, for the Results section, standardize the nomenclature for numbers below 10 in the manuscript. According to standard English writing conventions, numbers from one to nine (inclusive) should generally be spelled out rather than written as numerals. For example, instead of writing "2 haplotypes," you would write "two haplotypes".
Author Response
Comment 1: "I suggest including the new table (Table 4) immediately after the first paragraph in the Results section, renamed as Table 1."
Response 1: Thank you for the suggestion. We have followed your recommendation and moved the new table, now renamed as Table 1, immediately after the first paragraph in the Results section. The other tables were re-numbered accordingly.
Comment 2 "Lines 156-157: I suggest removing ' (table 1)' from the sentence 'The results of the study revealed a number of 635 different HLA-A/HLA-B/HLA-C/HLA-DRB1 haplotypes (table 1).'. Apply the same criteria to the sentences of lines 183 and 207."
Response 2: We appreciate your comment. We have removed the reference to “(table 1)” in lines 156-157, as well as in lines 183 and 207, in accordance with your suggestion.
Comment 3: "Table 4 does not show the HLA-DRB1 alleles. Please confirm."
Response 3: Thank you for pointing this out. We acknowledge that the HLA-DRB1 alleles were inadvertently omitted from Table 4. We have now added the HLA-DRB1 alleles to the table, ensuring that it fully reflects the intended data.
Comment 4: "In line 299, the haplotype does not include the HLA-C allele. Please confirm if it is correct."
Response 4: We appreciate your observation. We confirm that the haplotype in question does not include the HLA-C allele, the published data regarding only the Macedonian HLA-A/HLA-B/HLA-DRB1 haplotypes.
Comment 5: "In particular, for the Results section, standardize the nomenclature for numbers below 10 in the manuscript. According to standard English writing conventions, numbers from one to nine (inclusive) should generally be spelled out rather than written as numerals. For example, instead of writing '2 haplotypes,' you would write 'two haplotypes'."
Response 5: Thank you for your suggestion. We have carefully reviewed the manuscript and made the necessary changes, spelling out numbers below 10 in accordance with standard English writing conventions.
Reviewer 3 Report
Comments and Suggestions for Authors
Dear authors of this manuscript, I appreciate your efforts to address the suggestions made in the initial review of this study. I understand that it would be complicated and laborious to perform new statistical analyses related to quality control of the data used in this study without the availability of genotyping and/or whole-genome sequencing technologies to assess the ancestry of the samples and verify the identity of possible first-degree family relationships between them to declare that these are unrelated samples and that they belong to a homogeneous population labeled as the Romanian population. However, I assess that your study is a first approximation of the haplotype distribution in the HLA region, and I encourage you to design a new study that takes into account the suggestions made in the initial review to increase the study's statistical power
Author Response
Comment: "Dear authors of this manuscript, I appreciate your efforts to address the suggestions made in the initial review of this study. I understand that it would be complicated and laborious to perform new statistical analyses related to quality control of the data used in this study without the availability of genotyping and/or whole-genome sequencing technologies to assess the ancestry of the samples and verify the identity of possible first-degree family relationships between them to declare that these are unrelated samples and that they belong to a homogeneous population labeled as the Romanian population. However, I assess that your study is a first approximation of the haplotype distribution in the HLA region, and I encourage you to design a new study that takes into account the suggestions made in the initial review to increase the study's statistical power."
Response: Thank you for your thoughtful comments and constructive feedback, which all the authors really appreciate. We do consider your suggestions and will develop a future study aimed at enhancing the statistical power and addressing the outlined limitations.